# BioDiffusion: A Versatile Diffusion Model for Biomedical Signal Synthesis

**DOI:** 10.3390/bioengineering11040299

**Published:** 2024-03-22

**Authors:** Xiaomin Li, Mykhailo Sakevych, Gentry Atkinson, Vangelis Metsis

**Affiliations:** 1Department of Computer Science, Texas State University, San Marcos, TX 78666, USA; xminli@txstate.edu (X.L.); ukb12@txstate.edu (M.S.); 2Department of Computer Science, St. Edwards University, Austin, TX 78704, USA; gatkinso@stedwards.edu

**Keywords:** biomedical signal synthesis, generative AI, diffusion probabilistic model, deep learning, machine learning

## Abstract

Machine learning tasks involving biomedical signals frequently grapple with issues such as limited data availability, imbalanced datasets, labeling complexities, and the interference of measurement noise. These challenges often hinder the optimal training of machine learning algorithms. Addressing these concerns, we introduce BioDiffusion, a diffusion-based probabilistic model optimized for the synthesis of multivariate biomedical signals. BioDiffusion demonstrates excellence in producing high-fidelity, non-stationary, multivariate signals for a range of tasks including unconditional, label-conditional, and signal-conditional generation. Leveraging these synthesized signals offers a notable solution to the aforementioned challenges. Our research encompasses both qualitative and quantitative assessments of the synthesized data quality, underscoring its capacity to bolster accuracy in machine learning tasks tied to biomedical signals. Furthermore, when juxtaposed with current leading time-series generative models, empirical evidence suggests that BioDiffusion outperforms them in biomedical signal generation quality.

## 1. Introduction

Biomedical signal processing is significant across many common computing applications. The need for accurate, dependable data has been a driving force for innovations that have lead to improved assistive technologies and deeper insights into diagnostics, patient monitoring, and therapeutics. Electrocardiograms (ECGs), electroencephalograms (EEGs), and data from human activity sensors represent a treasure trove of information. Their analysis has ushered in transformative breakthroughs, but not without associated challenges.

One major hurdle faced in biomedical signal processing is the intricacies that arise due to limited dataset size, imbalances in datasets, artificial noise, and anomalies. These factors can critically compromise the performance of machine learning models, necessitating the development of innovative solutions. Historically, approaches like data augmentation, data resampling, and statistical analyses have been employed to mitigate these challenges. Yet, the inherently non-stationary and multivariate characteristics of biomedical signals add another layer of complexity. Encouragingly, recent research trends highlight an uptick in leveraging deep learning for enhancing the preprocessing of biomedical signals [1,2,3].

Deep learning, though powerful, is often constrained by the nuances of biomedical datasets. Recognizing these challenges, our study introduces the BioDiffusion model, a diffusion-based probabilistic approach tailored for biomedical signal generation. Designed to adeptly handle a plethora of generation tasks, BioDiffusion serves as a holistic solution to biomedical signal synthesis challenges. From expanding training dataset sizes to anomaly removal and super-resolution, our model’s adaptability offers a promising avenue for more efficient and precise analysis techniques in clinical applications.

Inspired by the Stable Diffusion model’s ability in image synthesis [4], we adapt the BioDiffusion model to work similarly with the unique traits of biomedical signals. To evaluate our model, we engage in a multi-faceted assessment, employing visual similarity comparisons, dimensionality reduction technologies like UMAP [5], and similarity scores such as wavelet coherence. Additionally, our research delves into the potential of synthesized signals for training new models, juxtaposing synthetic signals against real signals.

Through rigorous benchmarking against contemporary time-series synthesis models, our findings demonstrate the BioDiffusion model’s superior performance in generating high-fidelity biomedical signals. The implications of our proposed model are profound; it presents a significant stride toward enhancing diagnostics, improving patient monitoring, and advancing biomedical research.


**Main Contributions:**
Presentation of the BioDiffusion model, our innovative diffusion-based probabilistic approach tailored to address the complexities inherent in biomedical signal generation.Demonstration of our model’s versatility in handling diverse generation tasks, presenting a unified solution to biomedical signal synthesis.Comprehensive evaluation of the BioDiffusion model through both qualitative and quantitative metrics, underscoring its effectiveness and precision.Comparative analysis highlighting the superior capability of BioDiffusion in biomedical signal synthesis relative to existing state-of-the-art models.


The remainder of this paper is structured as follows: Section 2 delves into pertinent works related to signal synthesis. Section 3 provides an overview of Diffusion Probabilistic Models. Section 4 details our methodologies, including the development of BioDiffusion models and the evaluation metrics employed. Section 5 describes our experimental setup, the datasets used, and conducts a comparative analysis highlighting BioDiffusion’s superior performance. Section 6 addresses the significance, advantages, limitations, and future directions of our work. Finally, Section 7 concludes the paper. The source code can be found via the following link: https://github.com/imics-lab/biodiffusion (accessed on 5 March 2024).

## 2. Related Work

This section catalogs the pertinent literature in the fields of generative models for signal synthesis, particularly those using diffusion. Our objective is to offer a comprehensive perspective on their evolution, strengths, and constraints, especially in the context of time-series signal synthesis.

### 2.1. Generative Models in Signal Synthesis

Generative models aim to discern the inherent structure of data, enabling the generation of new samples mirroring the original data. Several paradigmatic approaches within generative models for time-series synthesis include:Generative Adversarial Networks (GANs): Composed of two adversarial networks— the generator and the discriminator—GANs aim for the generator to improve its synthetic data samples to deceive the discriminator. Their capabilities extend to various data types including time-series signals. Notable implementations include the transformer-based GAN by Xiaomin L. et al. [6] which sets a benchmark for synthetic time-series signal fidelity, TimeGAN by Jinsung Y. et al. [7] tailoring GANs for realistic time-series data, and the Recurrent Conditional GAN (RCGAN) by Cristóbal E. et al. [8] for time-series generation. Despite their proficiency in crafting realistic samples, GANs can exhibit training instability and suffer from mode collapse.Variational Autoencoders (VAEs): VAEs, through their encoder–decoder architecture, learn a probabilistic representation of data. Works such as that by Vincent F. et al. [9] exploit VAEs for imputing missing multivariate time-series values, while Fu et al. [10] leverage VAEs for augmenting time-series in human activity recognition. VAEs offer more consistent training than GANs but may produce less diverse samples, contingent on latent space distribution choices.Autoregressive Models: These models sequentially generate samples, with each new element contingent on prior elements. WaveNet by Aaron van den Oord et al. [11] exemplifies this, producing raw audio waveforms using dilated causal convolutions for long-range temporal relationship capture. Although proficient in modeling temporal dynamics, their sequential nature can be computationally slow and may falter in grasping extended dependencies.Other generative paradigms like Normalizing Flows, Restricted Boltzmann Machines, and Non-negative Matrix Factorization have been explored. However, their efficacy diminishes with multidimensional non-stationary time-series signals.

### 2.2. Diffusion Models for Time-Series Synthesis

Diffusion models harness latent variables to understand a dataset by modeling data point propagation through a latent space. They function by adding Gaussian noise to training data (forward diffusion) and subsequently reversing this process (reverse diffusion) to retrieve the data [12]. Their utility has been demonstrated in diverse arenas like image synthesis and molecule design [13].

Several prominent studies in diffusion models include:Yang L. et al.’s comprehensive discourse on deep learning-based diffusion models and their applicability to time-series tasks [12].Garnier O. et al. augmenting diffusion models for infinite-dimensional spaces, targeting audio signals and time series [14].Kong et al.’s exploration into audio synthesis through diffusion models [15] and Tashiro et al.’s venture into time-series imputation [16].Alcaraz et al.’s pursuit of time-series forecasting using diffusion models [17].

While these studies accentuate the capabilities of generative and diffusion models for time-series synthesis, challenges remain in terms of scalability, stability, and fidelity, especially for intricate biomedical signals. Our proposed BioDiffusion model stands as an endeavor to surmount these challenges, deriving inspiration from prior works while innovating for enhanced versatility and efficacy in biomedical signal synthesis. The forthcoming section elucidates the methodology underlying BioDiffusion, illustrating its potential to revolutionize biomedical signal synthesis.

## 3. Diffusion Probabilistic Models

This section provides an overview of the diffusion model theoretical foundations, detailing key components and processes. It explains the forward and backward processes, outlines the objectives of training a diffusion model, and describes how to incorporate conditions into the model training process.

Diffusion models [18,19] consist of a forward process that iteratively degrades data $x_{0}∼q\left(x_{0}\right)$ by adding Gaussian noise over *T* iterations:(1)$$\begin{array}{cc} q\left(x_{t}|x_{t-1}\right) & =N\left(x_{t};\sqrt{1-\beta_{t}}x_{t-1},\beta_{t}I\right), \end{array}$$(2)$$\begin{array}{cc} q\left(x_{1:T}|x_{0}\right) & =\prod_{t=1}^{T}q\left(x_{t}|x_{t-1}\right). \end{array}$$

The reverse process incrementally restores the noise-corrupted data:(3)$$\begin{array}{cc} p_{\theta }\left(x_{t-1}|x_{t}\right) & =N\left(x_{t-1};\mu_{\theta }\left(x_{t},t\right),\Sigma_{\theta }\left(x_{t},t\right)\right), \end{array}$$(4)$$\begin{array}{cc} p_{\theta }\left(x_{0:T}\right) & =p\left(x_{T}\right)\prod_{t=1}^{T}p_{\theta }\left(x_{t-1}|x_{t}\right). \end{array}$$

The forward process hyperparameters $\beta_{t}$ are set such that $x_{T}$ approximates a standard normal distribution. The reverse process optimizes the evidence lower bound (ELBO) [20], with the loss given by
(5)$$\;\begin{array}{cc} L_{\theta }\left(x_{0}\right) & =E_{q}\left[L_{T}\left(x_{0}\right)+\sum_{t>1}D_{\mathrm{KL}}\left(q\left(x_{t-1}|x_{t},x_{0}\right)\right.\right.\\ \; & \left.\left.∥p_{\theta }\left(x_{t-1}|x_{t}\right)\right)-\log p_{\theta }\left(x_{0}|x_{1}\right)\right], \end{array}$$
where $L_{T}\left(x_{0}\right)=D_{\mathrm{KL}}\left(q\left(x_{T}|x_{0}\right)∥p\left(x_{T}\right)\right)$.

Following prior work [18,19], the reverse process parameters are
(6)$$\begin{array}{cc} \mu_{\theta }\left(x_{t},t\right) & =\frac{1}{\sqrt{\alpha_{t}}}\left(x_{t}-\frac{\beta_{t}}{\sqrt{1-{\bar{\alpha }}_{t}}}\epsilon_{\theta }\left(x_{t},t\right)\right), \end{array}$$
(7)$$\begin{array}{cc} \Sigma_{\theta }^{ii}\left(x_{t},t\right) & =\exp\left(\log{\tilde{\beta }}_{t}+\left(\log\beta_{t}-\log{\tilde{\beta }}_{t}\right)v_{\theta }^{i}\left(x_{t},t\right)\right), \end{array}$$
with $\alpha_{t}=1-\beta_{t}$, ${\bar{\alpha }}_{t}=\prod_{s=1}^{t}\alpha_{s}$, and ${\tilde{\beta }}_{t}=\frac{1-{\bar{\alpha }}_{t-1}}{1-{\bar{\alpha }}_{t}}\beta_{t}$.

Improved sample quality is achieved by optimizing modified losses, resembling denoising score matching over multiple noise levels [19,21].

A critical aspect of diffusion models is the extension to conditional data generation, wherein both the data, $x_{0}$, and a set of conditions, c, are incorporated. The conditions can be any additional information or constraints provided externally, influencing the generative process. By assimilating c, the reverse process becomes
(8)$$\begin{array}{c} p_{\theta }\left(x_{t-1}|x_{t},c\right)=N\left(x_{t-1};\mu_{\theta }\left(x_{t},t,c\right),\Sigma_{\theta }\left(x_{t},t,c\right)\right) \end{array}$$

Intuitively, c offers an avenue to guide the generative model, providing a degree of control over the outputs. This inclusion makes diffusion models versatile, catering to scenarios like content-specific image generation or style-conditioned audio synthesis.

For the diffusion model architecture, we employ a feed-forward neural network (Section 4.4). It has distinct input layers for data, conditions c, and the time step. In line with the approach in [12], our model leverages multi-scale structures through convolutional layers, enabling the extraction of hierarchical information. The training strategy employs early stopping, hinging on validation set ELBO to prevent overfitting. Table 1 explains the notations used in the upper equations.

## 4. Methodology

In this section, we elaborate on the training and inference methodologies of the BioDiffusion model as employed in our study. Additionally, we describe the architecture of the diffusion model and the metrics implemented to validate the fidelity of synthetic data generated by the diffusion model.

### 4.1. Unconditional Diffusion Models

The unconditional diffusion model employs a Markov chain-based generation process, converting data iteratively between its original form and noise. This intricate transformation is portrayed in Figure 1.

**Forward Process**: Starting with the original signal, it is incrementally perturbed with Gaussian noise over a series of diffusion steps, spanning [0,T]. By the end of step *T*, the resulting signal retains the dimensions of the original but its data values adopt a normal distribution.

**Backward Process**: Initiating this process, signals derived from Gaussian noise serve as inputs at diffusion step *T*. As the model retraces the steps back to 0, it methodically diminishes the noise. Each step *t* consumes the previous step’s output (t+1) as its input. A crucial aspect during this phase is the evaluation of the Kullback-Leibler divergence (KL divergence) [22] between signals at the corresponding steps in both the forward and backward processes. The objective is to minimize this divergence. When the backward process culminates at step 0, the signals generated should closely mirror the original ones.

**Signal Generation**: Post training, the model is equipped to accept random Gaussian noise. By invoking the backward process, it can craft synthetic signals. This procedure is dubbed "unconditional" due to the absence of stipulations on the signal generation from the noise. Such a design empowers the diffusion model to assimilate the dataset’s entire distribution, endowing it with the capability to potentially produce any signal within the dataset’s feature space.

### 4.2. Label-Conditional Diffusion Models

Label-conditional diffusion models extend the framework of their unconditional counterparts by integrating scalar labels with each input datum. This inclusion of labels not only shapes the diffusion process but also allows for more targeted synthesis of signals, as elaborated below.

**Forward Process with Labels**: In this process, as depicted in Figure 2, original signals are systematically associated with their corresponding labels. Within the U-Net architecture (detailed in Section 4.4), each residual block is enriched with both the scalar label and the ongoing diffusion timestep, leveraging an embedding technique.

**Backward Process with Labels**: Here, the diffusion model ingests noise, drawn from a normal distribution, in tandem with a designated label. As the model progresses through the diffusion steps, there is a persistent focus on quantifying and minimizing the KL divergence between the signals emerging from the forward and backward processes.

**Synthetic Signal Generation**: The culmination of this methodology is a trained diffusion model possessing dual capabilities. It is not only attuned to the holistic data distribution of the dataset, but is also adept at crafting synthetic signals pertinent to a delineated class.

### 4.3. Signal-Conditional Diffusion Models

Signal-conditional diffusion models, visualized in Figure 3, introduce a nuanced methodology where signal conditions play a pivotal role exclusively during the backward diffusion phase, differentiating it from label-conditional approaches.

**Forward Process**: The forward diffusion process in the case of signal conditioning is the same as the original, unconditional diffusion.

**Backward Diffusion with Signal Conditioning**: For the backward phase, a perturbed signal forms the conditional input, which could stem from an original signal sample tainted by noise, artifacts, or even be a downsampled version mirroring the original signal’s dimensions. This conditional signal is amalgamated with noise drawn from a normal distribution. Following this combination, a convolutional layer refines it to align with the original signal’s structure. The remainder of the backward process strives to cleanse the noise and produce a clean signal resembling the original signal it was seeded with.

### 4.4. U-Net Architecture

The U-Net model depicted in Figure 4 is an encoder–decoder-type convolutional neural network architecture specifically designed for effectiveness in signal processing tasks. We modify the model depicted in work [23] to fit for time-series signals instead of N x N images. It features a symmetric structure with two primary pathways: the contraction path (encoder) and the expansion path (decoder). In Figure 4, the Down Sample block shows the encoder, and the Up Sample block shows the decoder.

The encoder consists of convolutional and max pooling layers that aim to capture the context within the input signal. This part of the network compresses the input, reducing its dimensionality to allow for the model to learn intrinsic patterns and features of the input data. The architecture comprises several blocks, each containing a convolutional operation followed by a residual block, which aids in learning an identity function and prevents degradation of network performance with increasing depth. Posterior to each residual block is an attention layer, which directs the model’s focus to the most salient features for reconstruction.

The decoder path expands the feature representation to precisely localize and reconstruct the signal. In U-Net, the up-sampling layers within the decoder increment the resolution of the output from the bottleneck. Subsequent to each up-sampling is a convolutional operation that constructs high-resolution features. A defining aspect of the U-Net is its skip connections that concatenate feature maps from the encoder to the decoder, integrating high-level and low-level features. This fusion allows for accurate localization by combining the general features from the contraction path with the detailed features in the expansion path.

In the diffusion model’s training and inference process, the U-Net underpins the architecture and is adapted to generate time-series signals. Signals at a given time step $x_{t}$ are concatenated with their corresponding time step embeddings and other conditional embeddings, such as low-quality signals or class labels, to provide the model with context for signal generation. These embeddings serve as conditions that direct the diffusion process towards generating the desired signal types.

During training, the U-Net learns to reverse the diffusion process by generating signals at time t−1 from those at time *t*, effectively learning to denoise signals. This iterative process is repeated from the final time step *T* to t=0, where the model generates a clean signal from one that has been fully diffused. This reverse iteration mirrors the forward diffusion process, enabling the model to reconstruct the original signal from its noisy counterpart and complete the U-Net’s training within the diffusion model framework.

### 4.5. Synthetic Sisnals Validation Metrics

In this section, we detail the metrics employed to validate the fidelity of the synthetic signals generated by our BioDiffusion model.

#### 4.5.1. Wavelet Coherence Score

Wavelet coherence is a statistical tool designed to assess whether two time series exhibit common oscillations at specific frequencies during a given time interval. It is calculated by taking the squared magnitude of the cross-wavelet spectrum and dividing it by the product of the power spectra of the individual signals. The resulting coherence values range from 0 to 1, with 1 indicating perfect coherence, signifying that the two signals are in complete synchrony at certain frequencies. This tool is particularly adept at analyzing non-stationary signals, where spectral content evolves over time. In our previous study [6], we modify this metric to measure the similarity between two sets of signals. We use the same method in this paper to compute the similarity between a set of real signals and a set of synthetic signals from the same category.

#### 4.5.2. Discriminative Score

Discriminative score is proposed in [7] as a way to quantitatively measure the similarity between sequences from the original and generated datasets. To accomplish this, the authors train a post hoc time-series classification model by optimizing a 2-layer LSTM to distinguish between the two datasets. In this method, each original sequence is labeled as real, while each generated sequence is labeled as not real. An off-the-shelf RNN classifier is trained to distinguish between the two classes as a standard supervised task. The classification error on the held-out test set is reported, which provides a quantitative assessment of the similarity between the two datasets.

#### 4.5.3. Umap Visualizations for Qualitative Signal Similarity Comparison

UMAP’s [5] approach to dimensionality reduction is rooted in manifold learning and topological data analysis. The algorithm begins by constructing a high-dimensional graph of the data, where each point is connected to its nearest neighbors in a way that reflects the local structure of the manifold. Then, UMAP optimizes the layout of this graph in lower-dimensional space using a force-directed layout approach, resulting in a projection that emphasizes the most important relationships and structures within the data.

We use the following steps to generate UMAP visualization plots and qualitatively compare the similarity between two sets of signals:**Preparation of Signal Data:** Flattening of both real set and synthetic set of signals into feature vectors.**Dimensionality Reduction:** Application of UMAP to reduce the high-dimensional feature space of each signal set to a two-dimensional (2D) embedding.**Visualization:** Plotting of the UMAP embeddings of both signal sets in the same coordinate system. Then, observation of the overlap and distribution of the two sets in the reduced space. Clusters of points from different sets that co-locate in the embedding space indicate a higher similarity.

#### 4.5.4. F1-Score for Imbalanced Dataset Classification Performance Evaluation

We use the F1-score to check the imbalanced dataset classification performance. The choice to utilize the F1-score rather than accuracy as the primary metric for comparing classification performance on imbalanced datasets is intentional and is grounded in the limitations of accuracy as a measure in such contexts. The F1-score is a statistical measure used to evaluate the accuracy of a binary classification model. It considers both the precision (*p*) and the recall (*r*) of the test to compute the score:$$p=\frac{TP}{TP+FP}$$
$$r=\frac{TP}{TP+FN}$$
where TP is the number of true positive results, FP is the number of false positive results, and FN is the number of false negative results.

The F1 score is the harmonic mean of precision and recall, which produces a single score that balances both by assigning equal weight to false positives and false negatives:$$F1=2\times \frac{p\times r}{p+r}$$
$$F1=\frac{2TP}{2TP+FP+FN}$$

By calculating and comparing the F1 scores for each class, we can gain a deeper understanding of the model’s performance across the entire range of classes, especially those that are underrepresented. This ensures a more robust and fair assessment of the model’s true predictive power.

## 5. Experimental Results

This section presents the various methodologies employed by our BioDiffusion models in the synthesis of biomedical signals. We partition our approach into three categories: unconditional, label-conditional, and signal-conditional diffusion processes. Our qualitative and quantitative evaluations underscore the efficacy of the generated synthetic data. We also benchmark our findings against state-of-the-art methods, underscoring the advantages of our model and pinpointing areas that need to be further developed. We aim to demonstrate that diffusion models are promising candidates for crafting high-caliber biomedical signals, potentially transforming myriad biomedical arenas.

### 5.1. Datasets

Our model’s performance is evaluated using three datasets: the Simulated dataset, the UniMiB human activity recognition (HAR) dataset [24], and the MIT-BIH Arrhythmia Database [25,26].

The Simulated dataset is a synthetic dataset with different signal patterns. These synthetic patterns are created using a combination of bell, funnel, and cylinder shapes. The dataset is generated for five classes, each with different characteristics, which are determined by their parameters. The parameters can be average amplitude, variance amplitude, variance pattern, etc. Each signal has 512 timesteps and one channel dimension. We can choose them to be any length and any dimension. Each class of signals in this dataset is evenly distributed. We use this dataset to test whether the diffusion model can learn the signal patterns properly before learning on more complicated, imbalanced real-world datasets.The UniMiB Dataset [24] is gathered using smartwatches; this dataset contains nine human activity classes with each signal capturing 151 timesteps across three acceleration dimensions. Adapted to our U-Net architecture, signals are resized to 128 timesteps. The training set contains 6055 samples, with class distributions that peak at 1572 and trough at 119 samples per class. The test set has 1524 samples, ranging from 32 to 413 samples per class, highlighting the dataset’s imbalance.The MIT-BIH Arrhythmia Dataset features 48 snippets of ambulatory ECG recordings spanning half an hour each from 47 subjects across five heart conditions [25,26]. The samples, originally recorded at 125 Hz, are adjusted to 144 in length for U-Net compatibility. The training set has 87,554 samples, with the majority class having 72,471 samples and the smallest class having 641. The test set includes 21,892 samples, ranging from 162 to 18118 samples per class, again underlining the dataset’s imbalance.

For an in-depth exploration of the datasets, refer to the Appendix A.

### 5.2. Visualization of Raw Signals

To assess the fidelity of synthetic signals visually, we present a comparative plot between several real and synthetic signals. For continuity, discrete signal values at each sampling interval are interconnected. Figure 5 illustrates a set of both real and synthetic signals derived from three distinct datasets. An immediate examination reveals the capability of our diffusion model in crafting synthetic signals that closely mirror the real signals.

### 5.3. Projection through Dimension Reduction

For each class in every dataset, an unconditional diffusion model is trained. The UMAP projection of synthetic signals in relation to the original ones for select data classes is depicted in Figure 6. Extended visualizations are accessible in the provided source code repository. When scrutinized, it becomes evident that even for signals of considerable length (e.g., 512 timesteps), our diffusion model adeptly recognizes and replicates the intricate signal patterns. Moreover, the synthetic signals span the entire feature spectrum inhabited by the genuine signals.

### 5.4. Similarity Scores

Using label conditions in the BioDiffusion model not only emulates the synthetic signal generation prowess of the unconditional model but also provides a guided synthesis tailored for specific classes. While the raw signals and UMAP projections closely resemble the ones in Figure 5 and Figure 6, the main advantage lies in the training efficiency. A singular label-conditional diffusion model suffices for a multi-class dataset, in contrast to the multiple models required by the unconditional counterpart for each class. Intriguingly, when it comes to sparsely represented data classes, the label-conditional model potentially outperforms the unconditional one. This edge is attributed to its capacity to generalize patterns across the dataset and utilize this knowledge for class-specific synthesis.

To underscore the fidelity of signals generated by our diffusion models, we calculate similarity scores across, wavelet coherence score and discriminative score, diverse signal classes. The results are cataloged in Table 2. Our BioDiffusion model’s outputs closely align with real signals, surpassing the fidelity of other similar techniques.

**Baseline Techniques**:C-RNN-GAN: A pioneering GAN-based solution for sequential data synthesis using two-layer LSTM for both generator and discriminator [27].RCWGAN: An enhanced version of C-RNN-GAN with conditional data input for controlled generation [8].TimeGAN: A groundbreaking GAN framework that harnesses a latent space for time-series synthesis, augmented with both supervised and unsupervised losses [7].SigCWGAN: Enhances the GAN process with conditional data and the Wasserstein loss for stable training [28].TTS-GAN: A novel transformer-centric GAN model focusing on high-fidelity single-class time-series generation [6].TTS-CGAN: An iterative version of TTS-GAN introducing a label-conditional transformer GAN, facilitating multi-class synthesis through a singular model [29].

### 5.5. Utility of Synthetic Signals in Addressing Class Imbalance

To explore the potential of synthetic signals in rectifying class imbalance issues, we constructed a classification experiment centered around the MIT-BIH dataset. This dataset, while demonstrating commendable overall accuracy, manifests stark class imbalances, often disadvantaging minority classes in terms of precision and recall.

**Experimental Setup:** Our initial step involved training a 1D-CNN classification model on the MIT-BIH Arrhythmia Dataset, a benchmark dataset in the field of cardiac signal analysis. During this phase, we observed performance discrepancies across different classes, particularly for minority classes, which exhibited lower precision and recall metrics. This was largely attributed to the dataset’s inherent imbalance where dominant classes overshadowed the minority classes, leading to a biased classifier.

To address this issue, we leveraged our developed BioDiffusion model, specifically using its label-conditional version. This model was used to generate synthetic signals that mirrored the characteristics of the underrepresented classes in the dataset. For instance, we generated additional synthetic signals corresponding to less frequent arrhythmia types such as ventricular ectopic beats (VEBs) and supraventricular ectopic beats (SVEBs), which typically have fewer examples in the dataset.

By incorporating these synthetic signals into the training set, we aimed to balance the class distribution and thereby reduce the bias towards the more prevalent classes. The addition of synthetic signals was carefully calibrated to ensure that the training set mirrored a more equitable class distribution, which was previously skewed.

Post generation, the identical 1D-CNN classification model was retrained with the new, balanced dataset. This model was then evaluated using the original, unchanged test set to provide an unbiased assessment of performance improvement. The results showed a marked increase in precision and recall for the minority classes, with the overall accuracy of the model also improving.

For a comprehensive evaluation, we compared our method’s performance with traditional resampling techniques such as random oversampling as well as other generative models that have been applied for signal synthesis in the literature. Our approach not only yielded an enhanced balance in class representation, but also improved the generalizability of the model, as evidenced by the performance metrics on the test set.

**Results and Analysis:** As presented in Table 3, synthetic signals crafted using our BioDiffusion model not only enhanced the training set, but also significantly bolstered the F1-score for the detection of minority classes. In contrast, signals synthesized by models like RCWGAN and C-RNN-GAN led the downstream classifier to a biased classification—predominantly towards the majority class (non-ectopic beats), effectively nullifying the F1-score for other classes. It is pivotal to note that during these evaluations, the real test set remained untouched and unseen throughout all generative model training phases.

### 5.6. Biodiffusion in Biomedical Signal Denoising, Imputation, and Upsampling

In time-series signal collections, three predominant noise types are frequently encountered: thermal noise, electrode contact noise, and motion artifacts. Thermal noise arises from the thermal agitation of electrons causing voltage or current fluctuations. Electrode contact noise stems from the changing electrical characteristics between electrodes and surfaces leading to signal baseline fluctuations. Motion artifacts, on the other hand, are sudden spikes in signals caused by physical disturbances like movement, unrelated to the actual biological activity being measured. These artifacts and noise types challenge the robustness of signal processing techniques. Leveraging BioDiffusion, we successfully denoised signals by taking the MIT-BIH dataset, adding artificial noise, and using it as an input for the diffusion model. Example results is shown in Figure 7.

Furthermore, BioDiffusion efficiently handles signal imputation tasks. Missing values in collected signals can be interpolated using our model, resulting in reconstructed signals that are impressively close to the original signals, as displayed in Figure 8.

Differing sampling rates across biomedical signals, due to equipment variations, necessitate resampling techniques. Traditional upsampling methods, while functional, often fail to capture intricate relationships among signal features. This problem is addressed with our signal-conditional diffusion model designed for signal upsampling, resulting in high-resolution signals that are almost indistinguishable from the originals.

A notable application of BioDiffusion lies in the generation of individualized signals. A scarcity of data samples from individual subjects can be a bottleneck for certain machine learning applications. However, our approach allows for a diffusion model to be trained on diverse signals, which is then fine-tuned using signals from an individual subject. This method generates synthetic signals that retain the unique patterns of the subject, enabling the expansion of subject-specific datasets.

For more visual examples of the output of BioDiffusion in upsampling and personalized signal generation, please see Appendix D.

## 6. Discussion

The introduction of the BioDiffusion model in this study represents a significant advancement in the field of biomedical signal synthesis. Our model is capable of generating novel instances of multi-channel biomedical signals, which is a notable enhancement over previous methodologies. It possesses the flexibility to be trained on datasets encompassing multiple classes, facilitating the application of transfer learning techniques across these classes. Furthermore, it offers the option to be label-conditioned, enabling the generation of instances belonging to a specific class during the inference phase. Additionally, the model can be conditioned on an existing signal, potentially containing noise or incomplete data, to produce a refined version of that signal. The capability of the BioDiffusion model to generate multi-channel signals is especially critical, filling a gap identified in the existing literature.

The challenge of creating synthetic data through machine learning models and utilizing these data to train new models is a question that extends its relevance beyond biomedical signal generation. In the context of Large Language Models (LLMs), where the demand for data by these models exceeds the creation of new human-generated content on the web, it remains uncertain whether synthetic data produced by LLMs can contribute to enhancing their own capabilities. Nonetheless, within the narrower scope of addressing class imbalance and improving signal quality, the generation of synthetic data has demonstrated its effectiveness.

In evaluating the quality of the signals generated by our model, we employed a combination of qualitative and quantitative methods previously outlined in the scholarly literature. However, it is important to acknowledge that current evaluation metrics are not flawless. Unlike images and text, the qualitative assessment of synthetic signal samples by humans is not straightforward, underscoring the imperative need for continued research into developing more robust quantitative metrics for this purpose.

Our findings indicate that as the complexity of the source signals increases, the model’s proficiency in generating high-quality synthetic counterparts decreases. This observation underscores that the endeavor to create high-fidelity synthetic biosignals is far from over. Future research should expand to encompass a broader spectrum of biomedical datasets, especially those characterized by greater complexity and synthesis challenges. Investigating a diverse array of model architectures and configurations, conducting sensitivity analyses to understand the impact of various hyper-parameters on the model’s generative capabilities, and tailoring model selection to specific signal characteristics are critical steps toward enhancing the quality of synthetic signals.

Lastly, while our research demonstrates the potential for advancing machine learning applications within the medical field, the area of biomedical signal synthesis remains underexplored within the generative AI landscape. To realize its full potential, it is crucial to encourage a greater number of researchers to delve into this domain. Collaborations with medical and biological experts are essential, as their expertise can significantly contribute to model refinement and the validation of synthetic data’s clinical relevance. Furthermore, increased funding and dedication to the open sharing of medical data, in compliance with ethical standards and privacy regulations, are imperative for fostering innovation and pushing the boundaries of the field forward.

## 7. Conclusions

In conclusion, the proposed BioDiffusion model is a novel and versatile probabilistic model specifically designed for generating synthetic biomedical signals. Our model offers a comprehensive solution for various generation tasks, including unconditional, label-conditional, and signal-conditional generation, which makes it a valuable tool for biomedical signal synthesis. We evaluated the quality of the generated signals using qualitative and quantitative assessments and demonstrated the effectiveness and accuracy of the BioDiffusion model in producing high-quality synthetic biomedical signals. Compared to state-of-the-art time-series synthesis models, our BioDiffusion model consistently outperforms its counterparts, showcasing its superiority and robustness in biomedical signal generation. The model’s versatility and adaptability have the potential to significantly contribute to the advancement of biomedical signal processing techniques, opening up new possibilities for improved research outcomes and clinical applications.

## Figures and Tables

**Figure 1 bioengineering-11-00299-f001:**
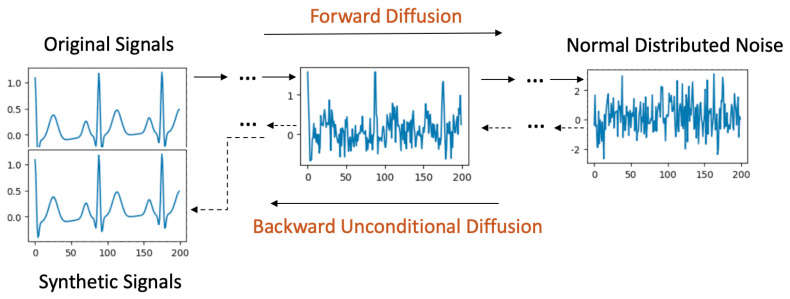
Unconditional Diffusion process.

**Figure 2 bioengineering-11-00299-f002:**
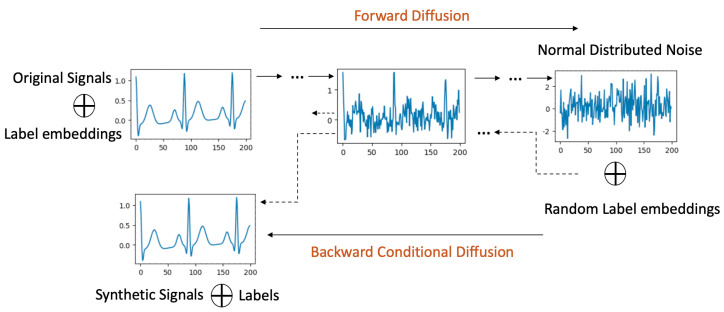
Label Conditional Diffusion process.

**Figure 3 bioengineering-11-00299-f003:**
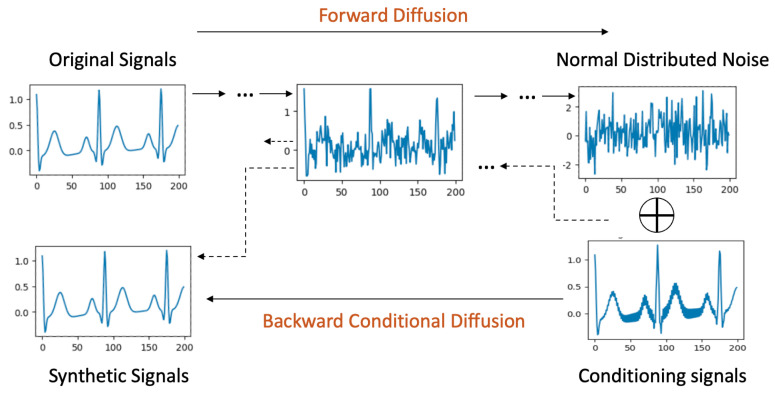
Signal Conditional Diffusion process.

**Figure 4 bioengineering-11-00299-f004:**
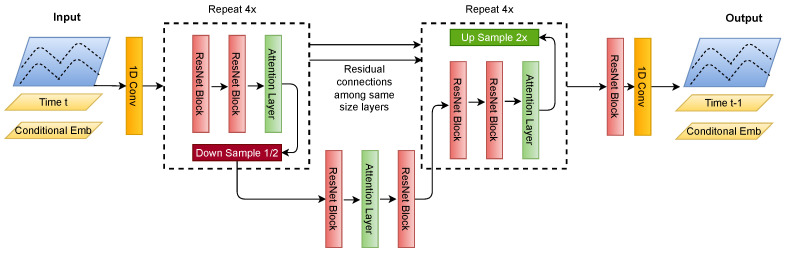
Description of the U-Net architecture for signals with skip connections.

**Figure 5 bioengineering-11-00299-f005:**
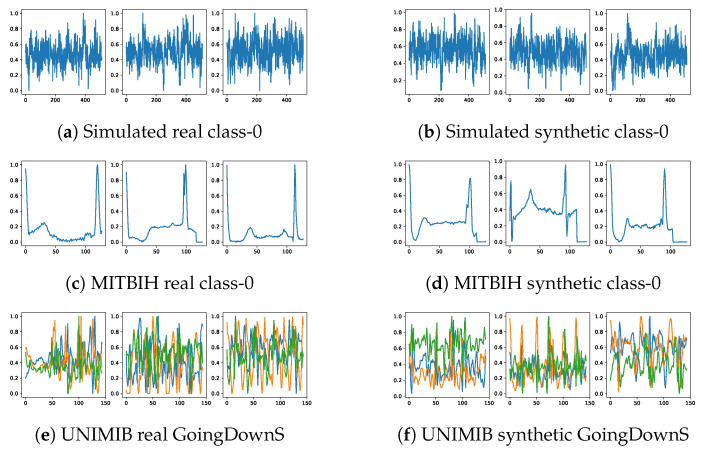
Raw signal comparison. Left column shows real raw signals. Right column shows synthetic raw signals generated by the BioDiffusion model.

**Figure 6 bioengineering-11-00299-f006:**
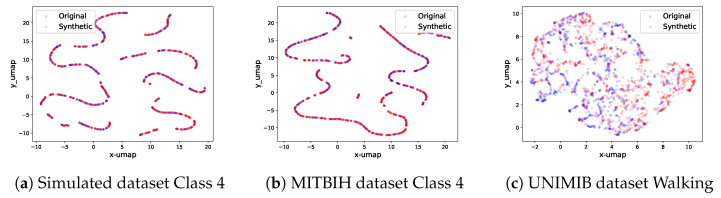
The real and synthetic data UMAP projection on three classes of three datasets. Each red dot represents one original signal after dimension reduction, whereas each blue dot represents one synthetic signal. From the plots, we can see that the sets of synthetic signals have similar distributions when compared to the sets of real signals in the 2D UMAP projection graphs.

**Figure 7 bioengineering-11-00299-f007:**
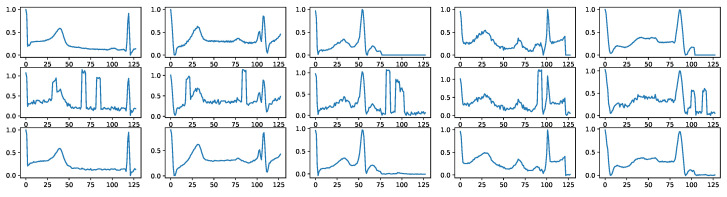
Example signal denoising results. First row: real signals. Second row: signals with noise. Third row: denoised signals using BioDiffusion.

**Figure 8 bioengineering-11-00299-f008:**
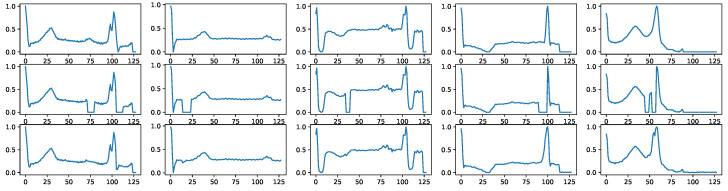
Example signal imputation results. First row: real signals. Second row: signals with random blanks. Third row: imputed signals by BioDiffusion model.

**Table 1 bioengineering-11-00299-t001:** Explanations of notation in diffusion model equations.

Symbol	Description
$x_{0}$	Original data point
*q*	Forward diffusion process transaction kernel
*T*	Total number of iterations in the forward process
*t*	Specific iteration step in the forward and reverse processes
$x_{t}$	Data point at iteration *t*
N(x;μ,Σ)	The general notation for the Gaussian distribution of *x* with the mean μ and the covariance Σ
$\beta_{t}$	Variance schedule parameter at iteration *t*
I	Identity matrix
*p*	Reverse diffusion process transaction kernel
$\mu (x_{t},t)$	Mean of the reverse process distribution at time *t*
$\Sigma (x_{t},t)$	Covariance of the reverse process distribution at time *t*
$L_{\theta }\left(x_{0}\right)$	Loss function for the diffusion model parameterized by θ
$E_{q}$	Expectation of the reverse process *q*
$D_{KL}$	Kullback–Leibler divergence
$\mu_{\theta }\left(x_{t},t\right)$	Mean of the reverse process at time *t*, parameterized by θ
$\Sigma_{\theta }^{ii}\left(x_{t},t\right)$	Diagonal covariance matrix of the reverse process at time *t*, parameterized by θ
$\alpha_{t}$	Variance accumulation parameter at iteration *t*
${\bar{\alpha }}_{t}$	Cumulative product of $1-\beta_{t}$ from time 1 to *t*
$\beta_{t}$	Adjusted variance schedule parameter at iteration *t*
*c*	Set of conditions or additional information provided externally
$p_{\theta }\left(x_{t-1}|x_{t},c\right)$	Reverse diffusion process of $x_{t-1}$ given $x_{t}$ and conditions *c*, parameterized by θ

**Table 2 bioengineering-11-00299-t002:** Comparison scores of real and synthetic data generated by different time-series generation models. The BioDiffusion model consistently achieves a higher Wavelet Coherence score and a lower Discriminative score in most instances, indicating that the synthetic signals it generates more closely resemble real signals compared to those produced by other baseline models.

Wavelet Coherence score (the higher the better)				
Models	SittingDown	Jumping	Non-Ectopic	FusionBeats
C-RNN-GAN	41.10	40.29	30.44	25.51
RCWGAN	39.90	38.85	29.72	22.97
TimeGAN	40.45	39.42	31.55	21.98
SigCWGAN	41.60	41.02	31.36	22.87
TTS-GAN	43.92	47.64	45.30	55.64
TTS-CGAN	45.07	47.64	47.79	58.34
**BioDiffusion**	**78.17**	**90.30**	**89.30**	**91.81**
Discriminative score (the lower the better)				
Models	SittingDown	Jumping	Non-Ectopic	FusionBeats
C-RNN-GAN	0.308	0.304	0.189	0.493
RCWGAN	0.294	0.311	0.483	0.499
TimeGAN	0.261	0.217	0.464	0.312
SigCWGAN	0.310	0.308	0.413	0.491
TTS-GAN	0.294	0.167	**0.107**	0.380
TTS-CGAN	0.191	**0.057**	0.162	0.261
**BioDiffusion**	**0.126**	0.121	0.159	**0.231**

**Table 3 bioengineering-11-00299-t003:** Per-class F1-scores for MIT-BIH classification using synthetic data to mitigate class imbalance. Abbreviations: N = Non-Ectopic Beats, A = Superventricular Ectopic Beats, V = Ventricular Beats, Q = Unknown Beats, F = Fusion Beats. The BioDiffusion model outperforms in most categories by achieving higher F-1 scores and secures the highest average score. This suggests that augmenting original imbalanced datasets with signals generated by the BioDiffusion model optimally enhances the classification F-1 scores compared to other generative models.

	N	A	V	Q	F	Average
Imbalanced	**0.97**	0.25	0.75	0.38	0.89	0.648
Re-sampling	0.50	0.65	0.64	0.81	0.85	0.69
TimeGAN	0.60	0.48	0.75	0.48	0.93	0.648
SigCWGAN	0.59	0.60	0.80	0.58	0.93	0.7
TTS-GAN	0.60	0.77	0.75	0.60	0.91	0.726
TTS-CGAN	0.66	0.78	0.77	0.85	0.93	0.798
**BioDiffusion**	0.73	**0.79**	**0.86**	**0.84**	**0.95**	**0.834**

## Data Availability

UNIMIB dataset link: https://www.mdpi.com/2076-3417/7/10/1101 MITBIH Dataset: https://physionet.org/content/mitdb/1.0.0/ (accessed on 5 March 2024).

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
