# Peer review of "BioDiffusion: A Versatile Diffusion Model for Biomedical Signal Synthesis"

_bioengineering, 2024, doi:10.3390/bioengineering11040299_

Round 1
Reviewer 1 Report
Comments and Suggestions for Authors
The manuscript is dedicated to the synthesis of biomedical signals via BioDiffusion method. The method could be very useful for augmenting of datasets and overcoming the imbalanced data problem. The manuscript is worth to be published, but only after a major revision in respect to the following questions, comments and recommendations:
1) Section ‘3. Diffusion Probabilistic Models’:
- The authors should provide explanation of the variables and abbreviations that are used for the presentation of equations from 1 to 8 (e.g. Dkl should be mentioned as Kullback–Leibler divergence at the place of its first appearance, etc.).
- The authors have written: A) “A crucial aspect during this phase is the evaluation of the KL divergence between signals at the corresponding steps in both the forward and backward processes. The objective is to minimize this divergence.”, and B) “As the model progresses through the diffusion steps, there’s a persistent focus on quantifying and minimizing the KL divergence between the signals emerging from the forward and backward processes.” From A and B it seems that the synthetic signals are quite similar to the original ones. How signals generated this way could enrich the database? Comments are necessary.
2) Section ‘4. Experimental Results’:
- Figure 5 – is there any correspondence between the illustrated “real” and the “synthetic” signals? Are the presented “real signals” the one that are used for generation of the “synthetic signals”? If yes, the difference is quite visible – e.g. the PQRST waveform is changed to a level that is not acceptable in case the signal label (annotation) would remain unchanged. Additional explanations on this topic are necessary.
- Figure 6 – In the caption it is written: “From the graphs, we can see that a set of synthetic signals are having similar distribution as a set of real signals in the 2D UMAP projections graphs.” This is not exactly the case with dataset UNIMIB.
- Subsection ‘4.4. Similarity Metrics and Results’ – Great part of the content of this section is not appropriate for section ‘4. Experimental Results’, but for ‘Method’ (such section is currently missing). Moreover, the authors should keep one and the same terminology – e.g. they address one and the same techniques as “cutting-edge” and “Baseline”, which should be avoided.
- Subsection ‘4.5. Utility of Synthetic Signals in Addressing Class Imbalance’ is again a mixture between methodology description and results. The methodological part should be addressed in more details in section ‘Methods’.
- Table 2 – Why F1-score is applied instead of accuracy for each class? Moreover, equations for the applied accuracy metrics should be provided in section ‘Methods’. I recommend the authors to use the standard annotation labels in MIT – i.e. N = Non-Ectopic Beats, A = Superventricular Ectopic Beats, V = Ventricular beat, Q = Unknown Beats, F = Fusion Beats.
3) Section ‘Discussion’ is missing.
Comments on the Quality of English LanguageModerate editing of English language required.
Reviewer 2 Report
Comments and Suggestions for Authors
Comment:
This manuscript introduces an optimized diffusion-based probabilistic model named BioDiffusion. It offers a versatile and comprehensive solution for generating synthetic biomedical signals, including unconditional, label-conditional, and signal-conditional generation tasks. Overall, the manuscript follows a logical and structured approach. In my opinion, this manuscript needs a minor revision to be published in this journal for the following reasons:
- In comparison to other models or empirical observations, how well does this model predict diffusion behavior?
- Why was sensitivity analysis not conducted to assess the robustness of the model to variations in input parameters?
- Which parameters influence the model outputs the most, and how does this affect the model's overall reliability?
In my opinion, this manuscript should be submitted after minor revision. Therefore, I recommend accepting this manuscript for publication in this journal.
Comments on the Quality of English LanguageModerate english modification is needed.
Reviewer 3 Report
Comments and Suggestions for Authors
In this paper the authors propose a diffusion model for biomedical signal synthesis.
The topic is very up to date and of general interest.
The document is well structured and written with good English.
The authors provide a good introduction, with very clear indications of the main contributions.
Section 2 and 3 provide a thorough overview of the supporting topics.
However, in subsection 3.4., the authors state “we modify…” while, in the beginning of section 3, the scope if “This section provides an overview of…”, creating a confusion between what is the authors’ work and what is a description of existent technologies. This must be clearly separated.
In addition, it seems that 3.4. cover an important part of the proposed approach, to which the authors have only dedicated a short paragraph. This paragraph also requires a better explanation of the procedure and should be part of a full definition of the used pipeline, as well as its inputs and outputs. I believe that a new section should be introduced, titled “BioDiffusion Model”, specifically dedicated to the explanation of the proposed pipeline.
Concerning the experimental results, the authors say “…pinpointing areas ripe for further refinement.”. In my personal opinion, the word “ripe” can be dubious and can lead the reader to think “if the are ripe (ready to use) then why they need refinement?”.
About the datasets, maybe they could be presented more clearly in a table, using as columns (suggestion) name, #records, #classes, #records/class, #sampling freq. The use of words like “timesteps”, “samples” and “snippets” are confusing. In a dataset context, “record” would be a better word. “samples” are the components of a discrete time signal. Why was a simulated dataset used? Can more information be provided about its generation? Is it available or the code to generate it?
Subsection 4.2 is named “…raw signals”. What are “raw” signals? The comparison should also cover frequency domain by showing the related spectrograms. (Time-domain observation provides very little information.) In addition, the authors should not claim simmilarity based on observation.
Subsection 4.3 mentions UMAP for the first and does not provide a reference neither an explanation for its usage or functionality in the context.
Subsection 4.4. mentions for the first time “The label-conditional…” without any prior introduction. This must be explained.
Concerning Table 1, how did the authors obtained values to calculate the baseline? Did the authors have implemented all the mentioned techniques?
The presented testing scenarios are interesting and cover a lot of possibilities. However they must be made clearer. The experimental setup in 4.5 contains very vague statements: “ to generate synthetic signals for each class. These synthetic signals were incorporated into the training set, striving to alleviate the dataset’s imbalance.”. What signals were generated? How imbalance was improved?
About the denoising scenario, what type of noise was included? How was the proposed system used to remove noise? Denoising effectiveness should be presented with objective metrics.
In conclusion, the supporting theory is good. However, despite the apparent good ideas, the authors miss many details about its explanation, requiring improvement in width and depth. The testing scenarios must be better organized and presented more clearly.
L198 provides some information about what the authors propose. Since
Round 2
Reviewer 1 Report
Comments and Suggestions for Authors
The authors have addressed all the questions and recommendations in my first report. In my opinion, the manuscript is suitable for publication in its present form.
Comments on the Quality of English LanguageMinor editing of English language required
Reviewer 3 Report
Comments and Suggestions for Authors
The authors have highly improved the paper quality while addressing my main concerns. The paper has now publishing quality.